# Individual subunits of a rhinovirus causing common cold exhibit largely different protein-RNA contact site conformations

Dieter Blaas [1✉]

Rhinoviruses cause the common cold. They are icosahedral, built from sixty copies each of the capsid proteins VP1 through VP4 arranged in a pseudo T = 1 lattice. This shell encases a ss(+) RNA genome. Three-D classification of single and oligomeric asymmetric units computationally excised from a 2.9 Å cryo-EM density map of rhinovirus A89, showed that VP4 and the N-terminal extension of VP1 adopt different conformations within the otherwise identical 3D-structures. Analysis of up to sixty classes of single subunits and of six classes of subunit dimers, trimers, and pentamers revealed different orientations of the amino acid residues at the interface with the RNA suggesting that local asymmetry is dictated by disparities of the interacting nucleotide sequences. The different conformations escape detection by 3-D structure determination of entire virions with the conformational heterogeneity being only indicated by low density. My results do not exclude that the RNA follows a conserved assembly mechanism, contacting most or all asymmetric units in a specific way. However, as suggested by the gradual loss of asymmetry with increasing oligomerization and the 3D-structure of entire virions reconstructed by using Euler angles selected in the classification of single subunits, RNA path and/or folding likely differ from virion to virion.

[1] Vienna Biocenter, Max Perutz Laboratories, Centre of Medical Biochemistry, Medical University of Vienna, Dr. Bohr Gasse 9/3, A-1030 Vienna, Austria.
✉email: dieter.blaas@meduniwien.ac.at

The packing of viral RNA genomes inside a protein shell is poorly understood; genome encapsulation might either involve active vectorial transport into the immature protein capsid by an ATP-driven motor, starting from the 5′-end that is first synthesised and thus first egresses from the replication complex[1], or by sequential co-assembly with viral building blocks, such as single subunits and/or pentamers[2]. Both models suggest that charge complementarity and/or stacking of nucleobases against aromatic amino acid residues might play a role in attachment of the RNA to the inner wall of the proteinaceous viral shell. Indeed, in X-ray structures of several picornaviruses unassigned electron density has been seen close to a conserved tryptophan in viral capsid protein 2 (VP2). It presumably originates from a stacked nucleotide[3–8].

The large family of picornaviruses includes the genus *ENTEROVIRUSES* with rhinovirus (RV) species A–C among many other animal and human pathogens so far classified into 47 genera (https://www.picornaviridae.com/). Although only occasionally giving rise to severe disease, RVs are economically important as they cause the common cold with its annoying and difficult-to-treat symptoms; currently no causal medication or vaccine is available[9]. In *ENTEROVIRUSES*, encapsidation depends on ongoing protein and RNA synthesis[10]. Since replication occurs on de novo formed membrane structures, the viral nucleic acid might be discriminated from cellular RNAs by local seclusion thus avoiding packaging of cellular RNA[11]. For the closely related Coxsackie and Polioviruses, the putative viral helicase 2C, a component of the replication complex, was shown to be involved in encapsidation by interacting with VP3, although the exact role of this association is unclear[12–14]. Regardless of the mechanism of the assembly process, in the mature virion, the RNA is presumably attached to the inner wall of the protein shell via multiple but weak interactions. Indeed, in Parechoviruses, another picornavirus genus, multiple, highly degenerate sequence stretches were found to interact with patches of the capsid proteins[15]. Because of the intrinsic asymmetry of the RNA, these contact sites might adopt unequal conformations and convey regionally limited asymmetric features to the otherwise strictly icosahedral particle[16]. Local asymmetry is also manifest during RNA exit occurring at one of the twofold or threefold axes of symmetry, as shown for Poliovirus[17,18] and suggested for RV-A2 by the geometry of attachment of subviral A-particles, intermediates of uncoating, to lipid membranes[19]. At least in RV-A2, the RNA leaves the virion with its 3′-end first, although it is unknown whether this can occur at any of the symmetry-equivalent positions or just at a predetermined site slightly deviating from perfect icosahedral symmetry[20,21]. One might speculate that the low numbers of VP0 that have not undergone maturation cleavage into VP4 and VP2 and are found in mature virions, play a directive role in assembly and/or directional RNA egress[22]. On the other hand, RNA exit in bulk through large holes opening via (reversible) dissociation of one or more pentamers from the virus shell has been proposed for the insect pathogen Triatoma virus[23] and the picornavirus Echovirus 18[24]. Such a dissociation/reassociation of subunits has also been purported for hepatitis B virus, where storage of two differently labelled virus populations for extended times showed the exchange of subunits suggesting that disassembly/reassembly was occurring[25]. The above asymmetric features of icosahedral viruses cannot be observed by X-ray crystallography and cryogenic electron microscopy (cryo-EM) employing icosahedral averaging. Even in non-averaged cryo-EM image reconstructions minor asymmetric features escape detection because the large symmetric parts might overwhelm the small differences during image alignment.

Avoiding the above pitfalls by 3D-classification of single asymmetric units of RV-A89 previously solved to 2.9 Å[26], such

local asymmetries became clearly visible. However, they gradually vanished upon extending the analysis onto dimers, trimers, and pentamers, suggesting that this local asymmetry does not go far beyond single subunits. This implies that the interaction of different sections of the viral RNA genome—adopting different secondary structures—with the N-terminal extensions of VP1, parts of VP4, and other amino acid residues at the protein interface, impart onto them different conformations. Assuming that the RNA establishes at least one contact with each asymmetric unit, my data indicate that it most probably does not follow the same assembly path in all virions but adopts different secondary structures within individual viral particles. A Hamiltonian path has been proposed to be followed by the genomic RNA in MS2 phage[27].

## Results

### Three-D classification of RV-A89 single asymmetric units reveals different conformations at the interface with the RNA genome.

In a previous study of the 3D-structure of a complex between RV-B5 and the capsid-binding pyrazolopyrimidine antiviral OBR-5-340, the non-binding RV-A89 was used as a control; its 3D-structure in the presence of OBR-5-340 was determined to 2.9 Å by imposing icosahedral symmetry[26], Fig. 1a. As expected from its lack of inhibition, OBR-5-340 was found to be absent from RV-A89 and the hydrophobic pocket in VP1, where most of such compounds bind, was empty. RV-A89 also lacked the natural pocket factor, a myristate occupying this pocket in many *ENTEROVIRUSES*[28]. Taken together, this led me

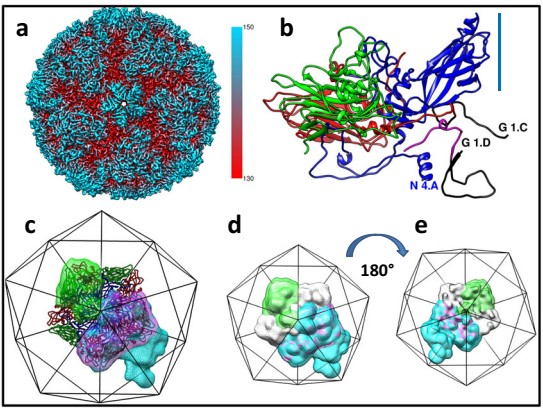

**Fig. 1 RV-A89 3D-structure and masks used for 3D-classification of subunits. a** Unsharpened density map of RV-A89 at 2.9 Å resolution (gold standard, 0.143 criterion) coloured from red to blue according to the distance in Å from the viral centre as indicated by the colour bar. Orientation I5, i.e., the view is down onto a fivefold axis of icosahedral symmetry. **b** Ribbon diagram of the atomic model of an asymmetric unit. VP1–VP4 are blue, green, red, and magenta, respectively; the view is sidewise onto the residues at or close to the N termini of VP1 (N 4.A), VP3 (G 1.C), and VP4 (G 1.D) and roughly parallel to a fivefold symmetry axis (blue line). Note that the RV-A89 map had several gaps in the density that were bridged by using the coordinates of RV-A16 (1AYN), black; the amino acid residues with low or insufficient density for reliable modelling were omitted in the deposited coordinates (6SK7). **c** External view onto a pentamer shown as ribbon diagram coloured as in (**b**); the approximate size and orientation of the entire virus is indicated with an icosahedron cage. The masks used for 3D classification of single subunits, dimers, and trimers are green, red, and blue, respectively. **d** External and **e** internal view (turned by 180° around the *y*-axis) of the masks as in (**c**) plus that of a pentamer in grey. The density levels of the respective masks were adjusted for best visibility. Note that the mask of the trimer (blue) includes the two subunits of the dimer (red). All panels created with UCSF Chimera (31).

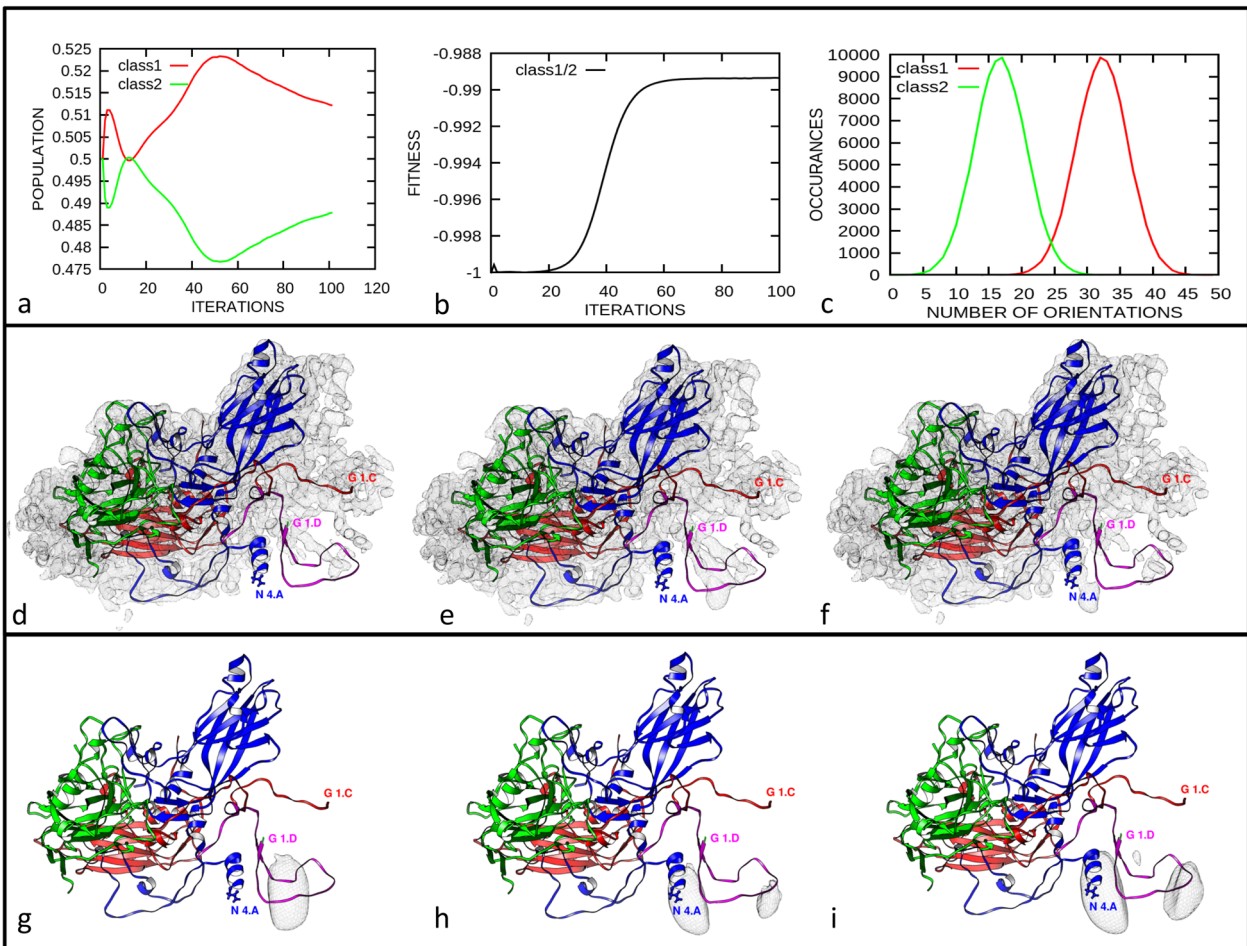

**Fig. 2 Three-dimensional classification without alignment of single asymmetric units of RV-A89 into two classes using the icosahedrally expanded cryo-EM dataset.** One asymmetric unit was used as reference map (Fig. 1b) and as a template for generating a corresponding mask (Fig. 1c–e, blue). As a control, the identical classification procedure was carried out with the number of 3D-classes set to one. **a** Evolution of the fraction of particle images segregating into the respective classes during the iterations. **b** Structural similarity, "fitness", as a function of iteration number. Note that a plateau, i.e., maximal difference was attained around iteration 60. **c** Numbers of orientations (out of the 60 angular combinations of each particle image obtained upon icosahedral expansion of the original dataset) segregating into class 1 and class 2, respectively. Maps of the control (**d** same classification, but the number of classes set to one), class 1 (**e**), and class 2 (**f**). Difference maps (depicted as mesh) of control and class 1 (**g**), control and class 2 (**h**), and of class 1 and class 2 (**i**). Positions of selected amino acid residues are indicated, last letter refers to the viral capsid protein, i.e., A = VP1, C = VP3, and D = VP4. The ribbon diagram is derived from the atomic coordinates from ref. [26] with gaps filled by using the coordinates of RV A16 (1AYN); see Legend to Fig. 1. The models were arranged for best viewing the N-terminal extensions of the VPs.

to consider this RV-A89 3D-structure as native and use it for the present investigation.

Using the script "relion_particle_symmetry_expand" implemented in Relion[29,30] I icosahedrally expanded the Euler angles of the aligned particle images obtained from the above analysis[26], resulting in a dataset of 5,851,680 particle orientations with sixty sets of Euler angles for each individual particle image compatible with the recorded projection image. By using the previously fitted atomic coordinates (pdb-6SK7), in which gaps were filled with the corresponding backbone coordinates of RV-A16 (1AYN), whose X-ray structure is more complete as it includes density for almost all amino acid residues[4] Fig. 1b, I excised a single asymmetric unit from the RV-A89 density map with Chimera[31]. The resulting map was then used as reference volume and for computing a soft mask sufficiently extended to include some density from residues contributed by the adjacent symmetry-related asymmetric units (green in Fig. 1c–e). The above dataset of particle images with the 60 equivalent orientations was then classified into two 3D-classes without alignment using Relion-3.0. The two 3D classes remained almost equally populated with increasing number of iterations

only showing a small fluctuation around iteration 10 (Fig. 2a). The former can be taken to suggest the presence of at least two distinct conformations represented to a similar extent. However, as also indicated by the disorder of parts of VP4 manifesting in low density in the 3D map of RV-A89 (map EMD-10222 and fitted atomic coordinates PDB-6SK7), many intermediate conformations might exist that segregate into one or the other class because of marginal similarities, suggesting 'an ordering function' of the RNA. The apparent resolution [maximum 4.0 Å; no goldstandard Fourier shell correlation procedure[29]] reported by Relion-3.0 plateaued around iteration 50. Note that it is certainly overestimated because of overfitting resulting from the regularisation parameter T set to 10 for better class separation.

I then determined a "similarity score" implemented in the xmipp programme suite as 'fitness'[32]; this score essentially represents the Pearson correlation between two volumes. The value is negative because it minimises the minus correlation. The higher its numerical value, the lower is the similarity between the two maps. Figure 2b depicts this 'fitness' between the two maps as a function of iteration number. Its value (−1 for identity at

iteration 0) started increasing at iteration 20 and attained a plateau around iteration 60. I discontinued the computation at iteration 100 although the distribution of the two populations appeared to still change slightly with the number of particle images partitioning into the two classes going again towards equity (Fig. 2a). It is of note that the number of angular orientations stemming from an identical particle image and segregating into one or the other class was quite different (Fig. 2c). Between 20 and 45 (peak at 32) of the 60 equivalent orientations compatible with a given particle image were contributing to class 1 and between 4 and 30 orientations (peak at 17) to class 2. This might indicate that more of the subunits exhibit conformations similar to that of class 1 as compared to that of class 2. The final maps resulting from these classifications superimposed onto the atomic model (Fig. 1b) rendered as ribbon drawing are shown as grey mesh in Fig. 2e (class 1) and 2f (class 2). The control, i.e., the map computed with identical parameters but with the number of classes set to one is depicted in Fig. 2d. Whereas the overall density distribution is virtually identical in the three maps, there are substantial differences at the capsid interior, in particular at the amino termini of VP1 (blue helix), and in the large loop of VP4 (magenta) that are close to the RNA genome[33]. This becomes even clearer in the respective difference maps represented as grey mesh and shown for the control minus class 1 in Fig. 2g, the control minus class 2 in Fig. 2h, and class 1 minus class 2 in Fig. 2i. This first analysis demonstrates that the RNA-contacting VP4 and the N-terminal extension of VP1 adopt at least two substantially different conformations in individual asymmetric units of the virion.

**Classification into 6, 12, and 20 3D-classes of single viral asymmetric units increasingly accentuates the differences of the RV-A89 protein–RNA interfaces.** The above two 3D classes presumably reflect the most dissimilar conformations in single subunits of RV-A89. However, each of the maps is necessarily an average of similar and less populated structural variants because of the limited number of classes included in the analysis. For a better differentiation I thus arbitrarily extended the analysis onto 6, 12, and finally twenty classes. At iteration 100, the nominal resolution for 6 classes (no gold-standard and skewed by over-fitting, see above) reported by Relion-3.0 was ~4.3 Å. The upper panel of Supplementary Fig. 1a shows the corresponding maps coloured according to "volume data gradient norm"; this presentation best emphasised the structural differences at the pro-tein/RNA interfaces and allows appreciating that the reminder of the density is again virtually identical in all classes. It is of note that the yellow to red blobs include density from the protein as well as from associated RNA. For orientation, the asymmetric unit, cut out from the original map (EMD-10222), and the cor-responding atomic model (pdb-6SK7, with the gaps filled with the coordinates from RV-A16, see above, depicted as a ribbon dia-gram), are included ("C" and "OM", respectively). The lower panel of Supplementary Fig. 1a shows the maps rendered as mesh enclosing the ribbon diagram of the atomic model above. Sup-plementary Fig. 1b depicts the evolution of the "fitness" of the fifteen pairs of the six density maps as function of iteration number. It indicates the largest structural divergence (i.e., the highest value in this particular presentation) of class 1 and class 5 and of class 5 and class 6 between iteration 60 and 100. For better appreciation of the differences the three maps above are sepa-rately presented in Fig. 3 together with the corresponding plot of the "fitness" parameter. Supplementary Figure 5 depicts zoom-in views of the above classes emphasising the large differences of the maps corresponding to VP1 and VP4 at the RNA–protein interfaces.

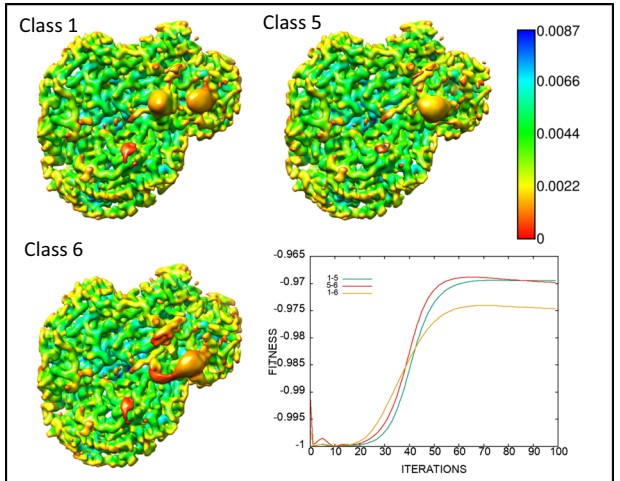

**Fig. 3 Three-dimensional classification of single asymmetric units of RV-A89 into six classes boosts the differences between the individual 3D-classes at the RNA/protein interfaces.** The three most divergent 3D-classes are shown together with a plot indicating their divergence as a function of the iteration number. Rendered with Chimera at sigma = 10 and coloured according to "volume data gradient norm" as indicated with the colour bar (for all six classes see Supplementary Fig. 1A, B). Maps of classes 1–6 were uploaded to the EMDB as examples (EMD-11618, class 1; EMD-11619, class 2; EMD-11620, class 3; EMD-11621, class 4; EMD-11622, class 5; EMD-11623, class 6).

The maps obtained from the classifications into twelve and twenty 3D-classes are shown in Supplementary Figs. 2 and 3A. The nominal apparent resolution was 4.6 and 4.9 Å, respectively. Taken together, the heterogeneity of the conformations at the protein/RNA interface becomes more evident with increasing number of classes; the data suggested the presence of at least 20, but most likely of more distinct conformations of the protein/RNA interaction sites. By visual inspection some single subunit classes appeared more similar to each other (see Supplementary Fig. 3a, e.g., class 4 and 20) than others (e.g., class 3 and 14). It is of note that one patch of low density gradient steepness (white arrow in Supplementary Fig. 3a, class 1) was present throughout all maps. It might correspond to a particularly strong and conserved interaction with some kind of putative packaging signal as suggested for the bacteriophages MS2 and GA[34]. Without signal subtraction and cutting out the asymmetric unit, hardware constraints did not allow extension onto more than 20 classes. However, upon signal subtraction ("Particle Subtraction" in Relion-3.1-beta), essentially the same results were obtained for up to 20 classes. Furthermore, in combination with a reduction of the box size from 450 px to 150 px, analysis of even 60 classes became possible albeit the apparent resolution decreased to below 6 Å for 12 of the classes (overall range: 3.9–7.3 Å). This extensive classification also revealed differences in the main part of the asymmetric units (Supplementary Fig. 6); such differences were minimal in up to 20 classes (compare to Supplementary Figs. 1a, 2, and 3a). The reproducibility of the density differences makes it highly unlikely to just represent noise.

**Conformational differences of the protein–RNA interfaces gradually diminish from subunit monomers to dimers to tri-mers to pentamers.** Are the conformational differences of the protein/RNA interaction sites also detectable in subunit oligo-mers? If one given monomer class were preferentially paired with another given monomer class the conformational diversity of the RNA-binding sites would be reduced. If so, this might suggest an

extension of the conserved protein/RNA binding-patches beyond subunit monomers. It would result in the signal not changing much with the degree of oligomerization. In the absence of such preferred pairing, similarities would gradually vanish with the degree of oligomerization until becoming completely smeared out in the 60-mer, the entire virion. I thus arbitrarily selected two adjacent asymmetric units, ("2/5 of a pentamer" see Fig. 1c–e blue), and subjected these dimers to 3D-classification, essentially as for single asymmetric units explained above. All six classes were almost equally populated (~17% each at iteration 100), and the densities at the RNA interaction sites were relatively similar (Supplementary Fig. 4, D1–6). However, none of its two single-subunits could be easily related to a single subunit class (compare to Supplementary Fig. 4, M1–M6). This can be taken to indicate that protein/RNA interaction sites detected in monomers are not identically represented in the dimers. In other words, the number of combined subunit conformations is probably much higher for the dimers and might attain many more than the maximal sixty conformations for the monomers. As my limited analysis only considered 6 classes, the density at the protein/RNA interaction sites is certainly being spread out because of averaging. The increasing disorder as a consequence of oligomerization was also suggested from the result of extending the analysis to trimers and to pentamers (Supplementary Fig. 4, T1–6 and P1–6); the six pentamer classes appeared almost identical despite they were representing half of the 12 pentamers of the virion.

**Relating single subunit classes to asymmetry in entire virions**. The above results suggested that oligomerization of single sub-units into di-, tri-, and pentamers results in a gradual waning of the differences in the protein/RNA conformations among the classes. I thus wondered whether reconstruction of the entire virion by using the angular orientations selected during 3D-classification of single subunits would show any asymmetric features remote and different from the very subunit taken as a reference. Out of the twenty 3D-classes (Supplementary Fig. 3A), I arbitrarily selected class 9 because it is mostly present once only per virion (Supplementary Fig. 3B), class 3, because it is most different from all other classes (see heatmap of the fitness values of the 400 combinations of all 20 3D-classes in Supplementary Fig. 3C), and class 1, because it is represented most often within a single particle image, peaking at five times per virion (Supplementary Fig. 3B). Figure 4 shows a thick slice (~29 Å) through the centre of the reconstructed maps oriented as to also pass through the corresponding single subunit class. The difference map, in which the original volume obtained by icosahedral averaging (Fig. 1a) was subtracted, is also depicted. From this it becomes clear that orienting the viral density map on the basis of any of the three considered single subunit classes fails to show any asymmetric feature remote from the respective subunit. The existence of a similar RNA conformation within a sizeable population of virions is, therefore, highly unlikely.

## Discussion

I here show that the stretches of different sequence, and consequently of different secondary structure in the genomic RNA within the protein shell of rhinovirus-A89 impart distinct conformations onto parts of VP4, the N-terminal extension of VP1, and probably onto other RNA/protein contact sites in discrete symmetry-related asymmetric units of the virion. Whereas icosahedral averaging revealed disorder presenting as low density of various parts of VP4 and, to a lesser extent of VP3 (see PDB-6sk7 in Fig. 1b for these gaps), density not seen and/or differing in the symmetric reconstruction emerged upon classification of single

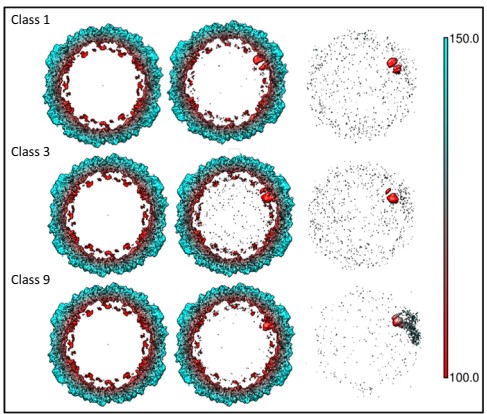

**Fig. 4 Three classes of single asymmetric units were selected, see boxed numbers in Supplementary Fig. 3A, from the 20 classes based on the below criteria.** Maps of the entire virus were reconstructed by using the Euler angles of these 3D-classes. Thick slices (29 Å wide) through the centre of the virion and oriented as to include the respective subunit class are shown in the middle panels. For comparison, the left panels depict the original volume obtained by icosahedral averaging (as in Fig. 1a). The right panels show the respective difference maps of the entire volumes (i.e., middle volume minus left volume). The 3D-classes were chosen on the basis of: Class (1) Maximal number of occurrences per particle image (peaking at five times; Supplementary Fig. 3B), Class (3) because of being most different from all others classes (see heatmap in Supplementary Fig. 3C), and Class (9) because of minimal number of occurrences per particle image (peaking at one time only; Supplementary Fig. 3B). The slices are coloured according to the distance from the centre given in Å as indicated in the colour key bar. Note that the difference map (in red) is shown in its entirety and not sliced!

subunits at the protein/RNA contact sites, whereas the rest of the protein remained invariant.

Analysis of the electron density of six classes of monomers, dimers, trimers, and pentamers of a single asymmetric unit computationally extracted from the cryo-EM map of the entire virion revealed a gradual loss of these differences with the degree of oligomerization (Supplementary Fig. 4). This is even more evident when the entire virion is considered; I selected three particular 3D-classes of single asymmetric units representing a particular conformation of its RNA/protein interaction site, removed the mask and reconstructed the entire virion from the Euler angles that had been selected in the classification. From this it became clear that the orientation of the selected subunit class is not correlated with a particular orientation of other subunits. In other words, there is no relationship with a specific conformation of any other subunit. This suggests that the path of the RNA from one to the next subunit is not conserved but rather differs in individual virions.

Viral assembly appears to require protein as well as RNA synthesis to occur in parallel; this might result in the 5′-vicinal stretches of the RNA first associating with building blocks of the protein shell, presumably a single asymmetric unit or a pentamer. This would be followed by interactions of the folding RNA with other asymmetric units. However, these latter ones are not necessarily in a unique spatial relation with the first one. My data appear to exclude that during assembly, the RNA first attaches to the subunits of a pentamer and then proceeds further to attach to the next pentamer. They rather suggest that each RNA molecule ends up inside a virion with different topology. This implies that the secondary structural elements of the RNA do not contact asymmetric units identically related to each other in individual virions. Current experiments are aimed at studying the above

questions by using biochemical methodology to reveal the secondary structure fold of viral RNA within the viral shell.

## Methods

**Cryo-EM structure determination of RV-A89.** All methods including RV-A89 preparation, purification, incubation with OBR-5-340, freezing, cryo-EM data collection, selection of full particle images, and image reconstruction of RV-A89 were detailed previously and can be found in the Supporting Information associated with ref. [26]. Since the capsid-binding antiviral OBR-5-340 does not inhibit infectivity of RV-A89 and no density attributable to this compound was found in the map, I considered it reasonable to assume that presence of OBR-5-340 and DMSO did not significantly perturb the native RV-A89 3D-structure. Therefore, the dataset of the aligned images (97,528) and the final unsharpened density map at 2.9 Å resolution were used in the present analyses.

Briefly, 9 µl RV-A89 at about 0.15 mg/ml in phosphate-buffered saline (PBS), were mixed with 1 µl of OBR-5-340 in DMSO to give final concentrations of 1 mM and 10% respectively, incubated at 37 °C for 1 h, n-dodecyl-ß-D-maltoside was added to a final concentration of 0.02%, and aliquots of 4 µl were applied to glow-discharged (60 s on carbon side, 25 mA) grids (Agar Scientific) with an additional layer of 4 nm continuous carbon. The grids were blotted for 3 s at 100% humidity and plunge-frozen in liquid ethane cooled with liquid nitrogen, by using a Vitrobot Mark III. Grids were imaged on a FEI Polara EM operating at 300 kV equipped with Gatan K2 Summit direct electron detector. Movies were recorded with a K2 camera in dose-fractionation mode at a calibrated magnification of 37,000×, corresponding to 0.97 Å per physical pixel. The dose rate on the specimen was set to 8 e/Å²s and total exposure time was 5 s, resulting in a total dose of 40 e/Å². Picking, image reconstruction and refinement were carried out with Relion-3.0-beta2[29,30] by using a previously determined map of RV-A2, Fourier-filtered to 60 Å, as starting volume. "Bad particles" were removed in repeated steps of 2D- and 3D-classifications and on the basis of excessive values of the relion parameter _rlnLogLikeliContribution. A model was built with the beta version of SwissModel https://swissmodel.expasy.org/interactive# (as of April 2018) allowing simultaneous modelling of hetero- oligomers using the Uniprot RV-A89 sequence B9V4A5_HRV89 and the automatically selected RV-A16 structure (1AYN) as template (the P1 region including the four capsid proteins has 74% identity and 84% positives). Coordinates were fitted and refined via Phenix[35] and Rosetta[36], and validated with Coot[37] and Molprobity[38]; the density map and the fitted atomic coordinates are available under EMD-10222 and PDB 6SK7, respectively. Note that parts of the VPs, in particular of VP4, could not be modelled without ambiguity and are thus omitted in 6SK7. However, for making the reference volumes and masks, missing parts were filled in with the more complete coordinates of RV-A16 (see Fig. 1b).

**3D-classifications.** By using Chimera[31], a zone extending by 5 Å over the above atomic coordinates was cut out from the refined, but not sharpened RV-A89 volume (EMD-10222) and used as starting map in the classifications and for computing the masks (filtered to 17 Å with an initial binarisation threshold of 0.001, extended by 1 px and with a soft edge of 10 px). Where explicitly mentioned, particle subtraction was performed with relion-3.1-beta prior to classification with essentially the same results; since the box size was reduced from 450 px to 150 px, classification into 60 classes became computationally possible (Supplementary Fig. 6). All 3D classifications were run for 100 iterations without alignment by using Relion-3.0 or 3.1-beta. The "degree of difference" among the conformations is expressed as the Pearson correlation between two aligned volumes as implemented in "xmipp_volume_align"[39]. This "fitness" has a negative value, i.e., −1 for identity and 0 for no similarity. Where indicated, controls were obtained by a 3D-classification run with one 3D-class only.

**Reconstruction of entire virions by using the alignment parameters from the selected 3D-classes.** Particle images with the associated alignment parameters taken from class 1, class 3, and class 9 from the run with 20 classes (Fig. 4) and chosen based onto the criteria described in the legend to Supplementary Fig. 3a were used in refinement runs by skipping alignment ("--skip_align" in Relion) and omitting masking. The final maps were sliced and oriented as to include the respective single subunit class with the most relevant densities. Note that the difference maps (the maps of the above classes minus the original map obtained by applying symmetry) are displayed in their entirety (not sliced) revealing all differences.

**Reporting summary.** Further information on research design is available in the Nature Research Reporting Summary linked to this article.

## Data availability

Maps of the six 3D-classes generated and analysed during the current study (and shown in Supplementary Fig. 1a) are available as examples in the Electron Microscopy Database repository (EMD-11618, class 1; EMD-11619, class 2; EMD-11620, class 3; EMD-11621, class 4; EMD-11622, class 5; EMD-11623, class 6). All other relevant data are available from the author upon reasonable request.

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

## Acknowledgements
Funded by the Austrian Science Fund (FWF), projects #27444 and #31392. Part of the calculations was carried out at the Vienna Scientific Cluster (project #71298); I am grateful to Siegfried Höfinger for helping with the submission parameters of the cluster and Carlos Oscar Sanchez Sorzano for drawing my attention to the xmipp implementation of the Pearson correlation. All cryo-EM data used in the presented analyses were taken from a previous study (ref. [26]).

## Author contributions
Dieter Blaas solely contributed to this work.

## Competing interests
The author declares no competing interests.
