## [Peer Review File · Communications Biology]

Reviewers' comments:

Reviewer #1 (Remarks to the Author):

RV structures, Blass: 2020

For many years, the vast majority of RV capsid structures were determined by crystallography or cryoEM techniques which for computational expediency, averaged the symmetry data into inferred commonalities for each of the 60 subunits. While functionally efficient, icosahedral averaging obscures most individual subunit peculiarities, especially those involved in internal RNA contacts. RNA packaging order, RNA egress and the random (?) distribution of uncleaved VP0 subunits are other mechanistic points that lose resolution when total symmetry is assumed.

This very nice study deeply data-mines RV-A89 cryo images at the 3D reconstruction level to assess the possibility of symmetry (or asymmetry) among individual protein:RNA contacts. The question is a difficult one and the Au here makes sequential multiple assumptions about possible outcomes, applying those parameters to the classification of images and extrapolating the outcomes. The procedures here are exactly those that should have been done and are well controlled for this type of analyses. A 2.9 cryo map was a wonderful starting resolution. The conclusion that each particle is probably unique in its contacts, solves a longstanding question in the field.

2 minor comments:

1. For those of us not so familiar with cryo-jargon for 3D reconstructions, a simple statement of approx distances being discussed would be helpful. For example, with the VP1/VP4 residues most likely to define these new structure classes, what is the approx range of modeled RMSD? A value, or range of values would give an inkling of how far these points really seem to "move." The figure visualizations make this a bit hard to estimate. E.g. in Fig 3 or S1A, do these classes differ by $<1\text{\AA}$ shifts somewhere or are they $>1-2\text{\AA}$?

2. Presumably, the original A89 dataset discarded images with no RNA density (i.e. empties), which for some RV can comprise up to 30% of the native particles on a grid. Does the new data predict the symmetry of every subunit in an empty becomes rearranged when it eventually makes RNA contact, or is there perhaps some intrinsic disorder in some bits of these proteins, "wiggle" if you will even before then? Do you guess, or can you speculate whether the RNA encounters different VP4-VP1 configurations and locks those in, or whether the specific, differential local RNA sequence perturbs these protein bits each into a unique conformation. Who do you think might be driving the obvious local induced fit? I'm not suggesting a parallel 3D classification of empties, just a comment or 2 on whether the RNA perhaps tames the innate wiggle of the interior proteins, or is the cause of the observed variation.

Reviewer #2 (Remarks to the Author):

Major comments

This manuscript uses cryoEM image processing methods to investigate RNA genome binding in rhinovirus capsids. I have several major concerns/areas which require significant clarification within this manuscript.

This manuscript and its conclusions are entirely based on cryoEM image processing, yet the methods are not presented in sufficient detail for the reader to be clear what analysis has been done. With this in mind, I present some comments but with the information provided it is entirely

possible that I may have misunderstood some of the analysis performed.

One of the main challenges here is that the results presented are a kind of negative result. The author reports very small differences in the locations of VP1 and VP4, but does not observe ordered density which one could confidently correspond to 'genome' in any of the EM density maps. 'Blobs' or low-resolution features can be seen in various classes on the genome-facing part of the capsid, which the author states could be attributed to protein or genome, but it is not convincing to me that these are not noise in the reconstruction. It is not clear that the lack of observation of a particular feature (eg ordered genome bases where it contacts VP) is due to it not being present, or limitations of the image processing methods.

In terms of the image processing, only one approach, namely symmetry expansion followed by classification without alignment, is performed. It is not clear to me why other common image processing techniques, such as focused classification on the region of interest (where viral proteins would contact the genome) were not applied. In addition, density subtraction to remove the contribution of the icosahedral protein capsid could have been applied. Finally, symmetry relaxation techniques could have provided additional insight. These approaches may have provided additional insight on top of the methods used here, and helped to add weight to the conclusion.

The final stage, where the author applies 'angular orientations selected during 3-D classification' and uses them to reconstruct a whole capsid- had so few details on the methods it makes no sense to me whatsoever in its current form. If you performed classification without alignment into 20 classes, those classes would have different particle subsets, but the angles would not have changed, because you performed classification without alignment? Do you mean that you reconstructed capsid using the particles from that class? Or did you perform an angular alignment elsewhere to obtain different sets of Euler angles? Far more details are needed on the methods in order to make an assessment about the conclusions.

The major conclusion of this paper is that the path of RNA from one subunit to the next varies in each individual virion. I think this is a slightly unremarkable conclusion, and I don't think I am aware of any publications advocating a single Hamiltonian path for the genome within rhinoviruses?

A specific major comment- L120- I don't know why the author has chosen to call the 'similarity score' 'fitness'? It seems to me like it's a similarity score- so should be labelled as such. Fitness has a very specific biological meaning and it is not clear to me why there is a link between similarity score and fitness, let alone a direct correlation. This requires significantly more explanation if the author wants to use the term 'fitness' for this metric.

I do agree with the author that the data presented convincingly shows that the VP1 and 4 that contacts the genome can adopt a range of different conformations, although these represent small variations. Unfortunately, due to the lack of detail around the image processing methods, and the fact it is more of a story about what you don't see rather than what you do see, I don't feel the data strongly supports the major conclusions of the manuscript (although I might agree with the conclusion!) in its current form.

Minor comments

Abstract and S3A says up to 20 classes but in text line 150 says 6,12 and 30 classes?

L54- 'and references therein' is not very clear- please refer the readers directly to the most relevant references

L101- what is the capsid binding antiviral (nucleic acid sequence, compound?)- assuming prior knowledge from ref 21.

L103- 'As expected, OBR-5-340 was absent from RV-A89 and the hydrophobic pocket in VP1 was empty.'- this sentence doesn't make sense to me? OBR-5-340 and RV-A89 are the two treatments? Should it read RV-B5 instead of RV-A89? I think the author is assuming some familiarity with previous work here.

L120- where ref 26 is cited, is the following a more appropriate citation? Vargas, J., Melero, R., Gómez-Blanco, J. et al. Quantitative analysis of 3D alignment quality: its impact on soft-validation, particle pruning and homogeneity analysis. Sci Rep 7, 6307 (2017). <https://doi.org/10.1038/s41598-017-06526-z>

L121- not very clear what 'higher number indicates less similarity' means in the context of a very small change of a negative number. This is better explained in the methods, might be worth stating that wording here.

L220- peculiar when I think you mean particular

Figure 1a- is it unsharpened density? Looks sharpened... (and no reason why you can't show sharpened for the figure- just state this)

Figure2- D,E,F- because identical atomic maps fitted, makes it hard to see the differences in density. Could the appearance of this be improved by fading the atomic map and highlighting in some way the main region of difference in EM density?

Figure S3A- reason for arrow not indicated in legend

Methods- The methods should be sufficient for the reader to replicate the experiment, and in this case does not contain sufficient detail of the methods used to generate the results, and needs significant expansion. The methods make no mention of the methods of icosohedral expansion of the particles mentioned in the text. There are no details about particle numbers used (or into each class). I would request a classification scheme to make it clearer to the reader how each map was derived.

Throughout three-D or 3D is used at different times- make consistent (3D would be my preference).

Where you state 'resolution', because of the methods use for the classification suggest you pre-fix with 'nominal resolution' (as you do in some places)

Acknowledgements- the author should acknowledge the source of the virus and cryoEM data used for this manuscript.

Data availability- there should be a section that points the reader towards relevant EMDB and PDB files. I would recommend that (at least some key classes) are uploaded to EMDB so readers can download the key files.

Reviewer #1 (Remarks to the Author):

RV structures, Blass: 2020

For many years, the vast majority of RV capsid structures were determined by crystallography or cryoEM techniques which for computational expediency, averaged the symmetry data into inferred commonalities for each of the 60 subunits. While functionally efficient, icosahedral averaging obscures most individual subunit peculiarities, especially those involved in internal RNA contacts. RNA packaging order, RNA egress and the random (?) distribution of uncleaved VP0 subunits are other mechanistic points that lose resolution when total symmetry is assumed.

This very nice study deeply data-mines RV-A89 cryo images at the 3D reconstruction level to assess the possibility of symmetry (or asymmetry) among individual protein:RNA contacts. The question is a difficult one and the Au here makes sequential multiple assumptions about possible outcomes, applying those parameters to the classification of images and extrapolating the outcomes. The procedures here are exactly those that should have been done and are well controlled for this type of analyses. A 2.9 cryo map was a wonderful starting resolution. The conclusion that each particle is probably unique in its contacts, solves a longstanding question in the field.

Thank you very much!

2 minor comments:

1. For those of us not so familiar with cryo-jargon for 3D reconstructions, a simple statement of approx distances being discussed would be helpful. For example, with the VP1/VP4 residues most likely to define these new structure classes, what is the approx range of modeled RMSD? A value, or range of values would give an inkling of how far these points really seem to “move.” The figure visualizations make this a bit hard to estimate. E.g. in Fig 3 or S1A, do these classes differ by <1Å shifts somewhere or are they >1-2 Å?

I did not attempt fitting the atomic coordinates into the densities of the distinct classes because, 1) some of these are including RNA density and, 2) the degree of disorder differs considerably manifesting in density lacking for residues in one class that might be seen in another class. This makes determining a reasonable RMSD quite difficult. However, in an attempt at complying with this reviewer’s request I now show more details of the three structurally most different subunits from the comparative analysis of six classes (shown in Fig. 3 and Fig. S1A, as also referred to by the reviewer). The zoom-in onto the relevant parts of the density, now shown in Fig. S5, allows roughly estimating the differences to about 2 to 3 Å (stated in the legend). Fig. S5 also hints at ways of how the residues might possibly be placed into the available densities. In the previous version of the manuscript I missed stating that several VP4 residues are disordered in the RV-A89 structure (see also modified Fig. 1B); the classification appears to separate this ensemble of conformations into the most frequent ones. This is also suggested from the classification into 60 classes (involving signal subtraction) now shown in Fig. S6.

2. Presumably, the original A89 dataset discarded images with no RNA density (i.e. empties), which for some RV can comprise up to 30% of the native particles on a grid. Does the new data predict the symmetry of every subunit in an empty becomes rearranged when it eventually makes RNA contact, or is there perhaps some intrinsic disorder in some bits of these proteins, "wiggle" if you will even before then? Do you guess, or can you speculate whether the RNA encounters different VP4-VP1 configurations and locks those in, or whether the specific, differential local RNA sequence perturbs these protein bits each into a unique conformation. Who do you think might be driving the obvious local induced fit? I'm not suggesting a parallel 3D classification of empties, just a comment or 2 on whether the RNA perhaps tames the innate wiggle of the interior proteins, or is the cause of the observed variation.

There was definitely a considerable fraction of empty particles on the micrographs and comparing single subunits of full and empty particles is an excellent suggestion! In fact, the amino-terminal extensions of the VPs were seen to be disordered in symmetric reconstructions of natural empty particles of RV-A2. I guess that this will be similar for empties of RV-A89. For VP4 it is more difficult to make a statement as it is either still connected to VP2 (in the natural empties) or lost (in the empty end-product of uncoating) see Pickl-Herk et al., Uncoating of common cold virus is preceded by RNA switching as determined by X-ray and cryo-EM analyses of the subviral A-particle. Proc Natl Acad Sci U S A 110, 20063-20068 (2013). I am currently determining the 3D-structure of the empty particles of RV-A89 and shall run such classifications in the future. In our native RV-A89 3D-structure there are parts with insufficient density for unbiased model building (see PDB-6SK7 and the now modified Fig. 1B). I thus added some words to draw the reader's attention to the fact that for the N-terminal extension of VP1 and parts of VP4, the contacts with the RNA appear to definitely impart increased order (see also the new Fig. S5, described above and L121ff and L162).

Reviewer #2 (Remarks to the Author):

Major comments

This manuscript uses cryoEM image processing methods to investigate RNA genome binding in rhinovirus capsids. I have several major concerns/areas which require significant clarification within this manuscript.

This manuscript and its conclusions are entirely based on cryoEM image processing, yet the methods are not presented in sufficient detail for the reader to be clear what analysis has been done. With this in mind, I present some comments but with the information provided it is entirely possible that I may have misunderstood some of the analysis performed.

Since the analyses presented in this manuscript were entirely carried out by using the previously collected cryo-EM data of RV-A89 and the derived refined map by routine image reconstruction methods [Wald et al., Cryo-EM structure of pleconaril-resistant rhinovirus-B5 complexed to the antiviral OBR-5-340 reveals unexpected binding site. Proc Natl Acad Sci U S A 116, 19109-19115 (2019)], I originally avoided overloading the manuscript by copying the Materials and Methods section of the above paper. However, from recurrent statements of this referee I understand that much of the methodology was unclear and insufficiently described. To improve clarity, I have now added an abbreviated version of the Methods chapter taken from the above reference and refer for more detailed methods to this very publication. I also added a number of missing details of the

classification procedure (see Methods).

One of the main challenges here is that the results presented are a kind of negative result. The author reports very small differences in the locations of VP1 and VP4, but does not observe ordered density which one could confidently correspond to 'genome' in any of the EM density maps. 'Blobs' or low-resolution features can be seen in various classes on the genome-facing part of the capsid, which the author states could be attributed to protein or genome, but it is not convincing to me that these are not noise in the reconstruction. It is not clear that the lack of observation of a particular feature (eg ordered genome bases where it contacts VP) is due to it not being present, or limitations of the image processing methods.

I perfectly agree with this reviewer in that it would be enlightening to identify density strictly attributable to ordered bases. However, despite careful analysis no clear pattern corresponding to nucleobases did emerge. This lack of well-defined density might indicate that a low degree of disorder and/or slightly different conformations are still present, even in the individual classes. Noise can most probably be excluded as the density differences between the classes are very clear and reproducible (with the 'body' of the subunits essentially being invariant in up to 20 classes). Furthermore, I added Figure S6 showing the results of classification into sixty classes by using data that had undergone signal subtraction. The overall result is essentially the same, which also speaks for 'real' density and not noise and documents the reproducibility of the analyses. I added a line to stress this point (L189).

In terms of the image processing, only one approach, namely symmetry expansion followed by classification without alignment, is performed. It is not clear to me why other common image processing techniques, such as focused classification on the region of interest (where viral proteins would contact the genome) were not applied. In addition, density subtraction to remove the contribution of the icosahedral protein capsid could have been applied. Finally, symmetry relaxation techniques could have provided additional insight. These approaches may have provided additional insight on top of the methods used here, and helped to add weight to the conclusion.

I agree with this reviewer in that all the suggested techniques might be employed to stratify the results. However, focusing the classification onto smaller parts of the structure degrades the resolution very much. Therefore, I chose to concentrate onto a single subunit instead of onto just some few amino acid residues in contact with the RNA. This choice is clearly subjective and a compromise. One might try classifying many differently sized parts of a subunit to find the optimal parameters. The computational cost of such calculations is, however, very high. To comply with this referee I chose to just try one of the suggested methods, namely signal-subtraction from the particle images, which gave essentially the same result as classification of single subunits without signal subtraction and allowed extending the analysis onto sixty classes. I now mention this (L189 and 197ff) and added Fig. S6. I also tried subtraction of the constant parts of the capsid volume but, as discussed in the manuscript, the differences vanished with the degree of oligomerization of the subunits. This all might be taken to indicate the lack of a conserved Hamiltonian path.

The final stage, where the author applies 'angular orientations selected during 3-D classification' and uses them to reconstruct a whole capsid- had so few details on the methods it makes no sense to me whatsoever in its current form. If you performed classification without alignment into 20 classes, those classes would have different particle subsets, but the angles would not have changed, because

you performed classification without alignment? Do you mean that you reconstructed capsid using the particles from that class? Or did you perform an angular alignment elsewhere to obtain different sets of Euler angles? Far more details are needed on the methods in order to make an assessment about the conclusions.

I understand from this comment that I was not sufficiently clear in this point; this was even worsened by a confusing error in the legend to Fig. 4. I corrected this error and added a much more detailed methods section to clearly explain how the reconstruction of the entire capsid was made. In short, the images of the entire particles were used but everything except one selected single subunit was masked away. Classification thus focuses only onto one single subunit with its Euler angles referring to and taken from the entire virion projection image. Upon removal of the mask, the entire virion can be reconstructed based on these Euler angles selected in the classification of the single subunits. This is slightly different from a procedure where part of the virion is extracted with signal subtraction as the latter involves re-centring and re-boxing. As mentioned above I now also include data on the classification into 60 classes as obtained with this latter method (Fig. S6). See also above.

The major conclusion of this paper is that the path of RNA from one subunit to the next varies in each individual virion. I think this is a slightly unremarkable conclusion, and I don't think I am aware of any publications advocating a single Hamiltonian path for the genome within rhinoviruses?

I agree with this referee in that to my knowledge no such Hamiltonian path is described for rhinoviruses but in Dykeman et al. Packaging signals in two single-stranded RNA viruses imply a conserved assembly mechanism and geometry of the packaged genome. J Mol Biol 425, 3235-3249 (2013) presents data in favour of such a path for MS2 phage. I now cite this paper and made this point clearer in the text (L94 ff).

A specific major comment- L120- I don't know why the author has chosen to call the 'similarity score' 'fitness'? It seems to me like it's a similarity score- so should be labelled as such. Fitness has a very specific biological meaning and it is not clear to me why there is a link between similarity score and fitness, let alone a direct correlation. This requires significantly more explanation if the author wants to use the term 'fitness' for this metric.

The term 'fitness' in this context is not my invention but stems from a routine in xmipp: As stated in the Acknowledgements "I am grateful to Carlos Oscar Sanchez Sorzano for drawing my attention to the xmipp implementation of the Pearson correlation". The function xmipp_volume_align uses a parameter called 'fitness'. One might discuss whether this name is appropriate or rather misleading but I preferred sticking to the parameter name used in this xmipp routine and explaining it according to its author.

I do agree with the author that the data presented convincingly shows that the VP1 and 4 that contacts the genome can adopt a range of different conformations, although these represent small variations. Unfortunately, due to the lack of detail around the image processing methods, and the fact it is more of a story about what you don't see rather than what you do see, I don't feel the data strongly supports the major conclusions of the manuscript (although I might agree with the conclusion!) in its current form.

Again, I realize that the methods section on image processing methodology was insufficient. As repeatedly explained above, this section is now expanded in the modified manuscript (see Methods).

Minor comments

Abstract and S3A says up to 20 classes but in text line 150 says 6,12 and 30 classes?

This typo is corrected

L54- 'and references therein' is not very clear- please refer the readers directly to the most relevant references

All six relevant references are now explicitly listed (L52)

L101- what is the capsid binding antiviral (nucleic acid sequence, compound?)- assuming prior knowledge from ref 21.

In the field of picornaviruses, in particular with respect to the species Enteroviruses, many capsid-binding drug candidates have been described and their binding sites in a hydrophobic pocket in VP1 have been determined. This is amply explained in the reference pointing to this paper but is now also detailed in the current manuscript (L99 ff) .

L103- 'As expected, OBR-5-340 was absent from RV-A89 and the hydrophobic pocket in VP1 was empty.'- this sentence doesn't make sense to me? OBR-5-340 and RV-A89 are the two treatments? Should it read RV-B5 instead of RV-A89? I think the author is assuming some familiarity with previous work here.

I understand that this statement was unclear; OBR-5-340, a bioavailable pyrazolopyrimidine, is a relatively novel capsid-binding virus-neutralizing compound. It does bind and neutralize RV-B5 but not RV-A89. Therefore, in the referenced paper on the structure determination of the complex between OBR-5-340 and RV-B5 [Wald et al., Cryo-EM structure of pleconaril-resistant rhinovirus-B5 complexed to the antiviral OBR-5-340 reveals unexpected binding site. Proc Natl Acad Sci U S A 116, 19109-19115 (2019)] RV-A89 was used as a negative, non-binding control. The very same cryo-EM data previously collected from this virus were thus used here with the reasonable assumption that the 3D-structure of RV-A89 determined in the presence of this non-interacting compound was identical to that of native virus (see changes made at L102 ff and new chapters in Methods).

L120- where ref 26 is cited, is the following a more appropriate citation? Vargas, J., Melero, R., Gómez-Blanco, J. et al. Quantitative analysis of 3D alignment quality: its impact on soft-validation, particle pruning and homogeneity analysis. Sci Rep 7, 6307 (2017). <https://doi.org/10.1038/s41598-017-06526-z>

I was referring to the software package xmipp here but I entirely agree in that the above paper is important and it might be a good idea to validate the 3D maps with this methodology. Nevertheless, taking into account the number of different maps I believe that this would be an unwarrantedly complicated task. Since I did not use this method, I refrained from citing this paper.

L121- not very clear what 'higher number indicates less similarity' means in the context of a very small change of a negative number. This is better explained in the methods, might be worth stating that wording here.

I admit that this is somewhat confusing but it is exactly what the Pearson correlation is indicating here. I am citing the wording of one of the authors of xmipp (see also Acknowledgements) and above.

L220- peculiar when I think you mean particular

Yes, was changed

Figure 1a- is it unsharpened density? Looks sharpened... (and no reason why you can't show sharpened for the figure- just state this)

It is NOT sharpened, this was and is mentioned in the Figure legend

Figure2- D,E,F- because identical atomic maps fitted, makes it hard to see the differences in density. Could the appearance of this be improved by fading the atomic map and highlighting in some way the main region of difference in EM density?

I added Fig. S5 that now very clearly shows details of the differences. I also give a rough visual estimate of the differences (2 to 3 Å). See also my answer to reviewer 1.

Figure S3A- reason for arrow not indicated in legend

is now indicated

Methods- The methods should be sufficient for the reader to replicate the experiment, and in this case does not contain sufficient detail of the methods used to generate the results, and needs significant expansion. The methods make no mention of the methods of icosohedral expansion of the particles mentioned in the text. There are no details about particle numbers used (or into each class). I would request a classification scheme to make it clearer to the reader how each map was derived.

As already mentioned at several instances above, it has become clear to me that the entire description of the employed methodology was not sufficiently detailed. I have thus extended the Methods section by copying large passages from the referenced paper and adding details specific for the present work.

Throughout three-D or 3D is used at different times- make consistent (3D would be my preference).

I changed the only occurrence of 'three-D' in the text to '3D'. All other occurrences of 'Three-D' were at the beginning of a sentence, where, I believe, the use of numbers should be avoided.

Where you state 'resolution', because of the methods use for the classification suggest you pre-fix with 'nominal resolution' (as you do in some places)

Is now changed to 'apparent' or 'nominal' resolution

Acknowledgements- the author should acknowledge the source of the virus and cryoEM data used for this manuscript.

The data used for the present analyses were taken from previous collaborative work of my and other groups. I thus did not find it appropriate to thank myself for the virus (which is not even being used here for experimentation) nor for the data, as they were produced by me and my collaborators. However, the cryo-EM data were and are appropriately cited and they can be assessed in the EMDB. The reference to these data is now included in the Acknowledgements.

Data availability- there should be a section that points the reader towards relevant EMDB and PDB files. I would recommend that (at least some key classes) are uploaded to EMDB so readers can download the key files.

References to the relevant data were and are given in the text. I arbitrarily decided to upload the maps of classes 1, 5, and 6 from the classification into six classes upon acceptance for publication. This is now stated in the manuscript (L501).

REVIEWERS' COMMENTS:

Reviewer #1 (Remarks to the Author):

Revision has strengthened and clarified the data presented in this manuscript. Since the analyses are mostly computational on existing datasets, clarity in the methods chosen for application is especially important. Revision has not however, changed the essential conclusions, namely that individual RV protomers are largely unique in the manner in which they interact with encapsulated RNA. The findings push the dataset to the maximum, but such analyses do provide important information about the lack of innate symmetry inside the particles presumably as a function of the packaging or VP0 cleavage functions.

Thank you for new fig S5. It clarifies a lot of questions about just how much difference there is in the various classes. The changes are remarkably local, not wholesale rearrangements. Eventually, a comparison with uncleaved VP0 empties will be even more informative but those results will take a while, understandably.

Reviewer #2 (Remarks to the Author):

The manuscript is now clearer in terms of the image processing methods used, which is very helpful to the reader.

Just one comment- on reading the revised manuscript and with the comment around 'conserved assembly mechanisms' in the rebuttal letter makes me think the author is accidentally misdirecting the reader about what hypothesis they are trying to test, by talking about a conserved Hamiltonian path (eg a single or small number of paths that the genome will take through the capsid- as described for MS2), when I think he means conserved assembly mechanism/packaging signals (where you might observe a limited ordered density associated with specific asymmetric units). I think, as Hamiltonian paths have never ever been suggested for this virus type, that it would be better to remove references to Hamiltonian paths and instead say 'conserved assembly mechanism/packaging signals' where appropriate?

I still feel this manuscript provides something of a 'negative result' it is always more difficult to describe findings where you are showing the absence of something, but the description of the work carried out is now sufficient for the reader to be able to understand the analysis done.

Please see my answers in bold italics

Reviewer #1 (Remarks to the Author):

Revision has strengthened and clarified the data presented in this manuscript. Since the analyses are mostly computational on existing datasets, clarity in the methods chosen for application is especially important. Revision has not however, changed the essential conclusions, namely that individual RV protomers are largely unique in the manner in which they interact with encapsulated RNA. The findings push the dataset to the maximum, but such analyses do provide important information about the lack of innate symmetry inside the particles presumably as a function of the packaging or VPO cleavage functions.

Thank you for new fig S5. It clarifies a lot of questions about just how much difference there is in the various classes. The changes are remarkably local, not wholesale rearrangements. Eventually, a comparison with uncleaved VPO empties will be even more informative but those results will take a while, understandably.

Thank you!

Reviewer #2 (Remarks to the Author):

The manuscript is now clearer in terms of the image processing methods used, which is very helpful to the reader.

Just one comment- on reading the revised manuscript and with the comment around 'conserved assembly mechanisms' in the rebuttal letter makes me think the author is accidentally misdirecting the reader about what hypothesis they are trying to test, by talking about a conserved Hamiltonian path (eg a single or small number of paths that the genome will take through the capsid- as described for MS2), when I think he means conserved assembly mechanism/packaging signals (where you might observe a limited ordered density associated with specific asymmetric units). I think, as Hamiltonian paths have never ever been suggested for this virus type, that it would be better to remove references to Hamiltonian paths and instead say 'conserved assembly mechanism/packaging signals' where appropriate?

This is a good suggestion! I have made the respective changes

I still feel this manuscript provides something of a 'negative result' it is always more difficult to describe findings where you are showing the absence of something, but the description of the work carried out is now sufficient for the reader to be able to understand the analysis done.

Thank you!